# Polymer-Grafted 3D-Printed Material for Enzyme Immobilization—Designing a Smart Enzyme Carrier

Daniela Eixenberger [1,*] , Aditya Kumar [1], Saskia Klinger [1], Nico Scharnagl [2] , Ayad W. H. Dawood [1] and Andreas Liese [1,*]

1   Institute of Technical Biocatalysis, Hamburg University of Technology, 21073 Hamburg, Germany
2   Helmholtz-Zentrum Hereon GmbH, 21502 Geesthacht, Germany
*   Correspondence: daniela.eixenberger@tuhh.de (D.E.); liese@tuhh.de (A.L.); Tel.: +49-40-42878-3218 (A.L.)

**Abstract:** One way to enhance the flow properties of packed bed reactors, including efficient mass transfer and high catalyst conversion rates, is the use of 3D printing. By creating optimized structures that prevent channeling and high pressure drops, it is possible to achieve the desired target. Nevertheless, additively manufactured structures most often possess a limited surface-area-to-volume-ratio, especially as porous printed structures are not standardized yet. One way to achieve surface-enhanced 3D-printed structures is surface modification to introduce surface-initiated polymers. In addition, when stimuli-sensitive polymers are chosen, autonomous process control is prospective. The current publication deals with the application of surface-induced polymerization on 3D-printed structures with the subsequent application as an enzyme carrier. Surface-induced polymerization can easily increase the number of enzymes by a factor of six compared to the non-modified 3D-printed structure. In addition, the swelling behavior of polyacrylic acid is proven, even with immobilized enzymes, enabling *smart* reaction control. The maximum activity of Esterase 2 (Est2) from *Alicyclobacillus acidocaldarius* per g carrier, determined after 2 h of polymer synthesis, is 0.61 U/$g_{support}$. Furthermore, universal applicability is shown in aqueous and organic systems, applying an Est2 and *Candida antarctica* lipase B (CalB) catalyzed reaction and leaving space for improvement due to compatibility of the functionalization process and the here chosen organic solvent. Overall, no enzyme leaching is detectable, and process stability for at least five subsequent batches is ensured.

**Keywords:** additive manufacturing; surface-induced polymerization; stimuli-sensitive; polyacrylic acid; Esterase 2; CalB

## 1. Introduction

The immobilization of enzymes is one key aspect for the successful application of biocatalysis in industrial processes. The fixation of enzymes on a support material, the carrier, is essential for an easy recovery and re-use of enzymes, leading to simplified downstream processes. Additionally, immobilization can enhance the stability of the biocatalyst, making their application even more economic [1–3]. Factors affecting the performance of enzymes immobilized on a support include pore size, particle size, porosity, shape, chemical composition, and type of chemical bond [4]. Nevertheless, the surface-area-to-volume ratio is of utmost importance when targeting a high enzyme load in a restricted reaction volume. However, the final enzyme activity introduced into the reaction system depends on the amount of enzyme immobilized and, thus, on the porosity of the material, which in turn may limit mass transport [5]. In addition, smaller particles lead to an increase of pressure drop in a fixed-bed system [6].

Since randomly packed fixed-bed reactors suffer from inhomogeneous flow, where phenomena such as channeling affect the overall performance of the reactor [7], one way to respond flexibly is to fabricate 3D-printed carriers. Additive manufacturing is currently

applied using different methods, such as fused deposition modeling, selective laser sintering, or inkjet printing, among others [8]. Depending on the method, various materials, such as polymers, alloys, metals, or ceramics, can be selected [8]. The here used method is selective laser sintering. It belongs to the powder bed fusion techniques and applies a powder bed where the powder of each layer is fused together by a laser [8]. The excess powder is removed subsequently.

Overall, additive manufacturing is used to produce complex geometries in a fast and cost-effective manner [9]. In comparison to traditional subtractive manufacturing, additive manufacturing allows the production of small parts while saving energy and material resources [10]. This technique can bridge the gap between theory and experimental set-ups, as it can be used to reproducibly fabricate high-resolution structures that have previously been studied theoretically in computational fluid dynamics simulations [11]. Therefore, it is easily applicable to design fluid dynamically optimized structures to achieve outstanding target properties, such as a proper mixing quality in a two-phase system, e.g., for process intensification, as previously shown by Büscher et al. [12] and Spille et al. [13].

Additionally, 3D-printed matrices were already reported as immobilization support [14,15], where the increase in the surface-area-to-volume-ratio is limited to surface roughness. Without porous microstructures, the bottleneck of implementation is the limited surface-area-to-volume-ratio. Overall, to place additively manufactured structures as an competitive immobilization matrix to traditional carriers, surface enhancement is essential.

Porous 3D printing is not yet standardized and lacks information regarding the exact correlation between process control and the final 3D object [16]. Furthermore, Stoffregen et al. classified porous 3D printing methods in geometrically defined lattice structures porosity (GDLSP) and geometrically undefined porosity (GUP) [17]. GDLSP is finally restricted by the achievable resolution of the chosen 3D printing process, whereas GUP is designed to achieve varying pore sizes and distributions by applying different processing parameters [17]. Therefore, the synthesis of an additional layer of polymer or even surface-initiated polymer brushes that lead to an increased surface-area-to-volume-ratio are chosen here to achieve higher surface-area-to-volume-ratios. A further advantage of the application of surface-initiated polymerization is the usage of stimuli-sensitive material. These materials can achieve both *smart* and surface-enhanced enzyme carriers, as stimuli-sensitive material undergoes a macroscopic change such as swelling or shrinking when triggered by an external stimulus, e.g., temperature, pH, or substrate concentration [18]. These properties make autonomous process control by application of stimuli-sensitive material prospective.

Enzymes were already successfully immobilized on stimuli-sensitive polymers [19–21], such as polyacrylic acid (PAA) [22,23], poly(2-(diethylamino)ethyl methacrylate [24], and poly(*N*-isopropylacrylamide) [23]. Nevertheless, previous publications did not focus on the application of modified 3D-printed scaffolds as an immobilization matrix. To the best of our knowledge when these materials have been used for enzyme immobilization, no stimuli-sensitive behavior has been studied to date. Even though the post-processing of 3D-printed surfaces is the focus of research, especially studies tackling the optimization of surface properties, so far, polymerization has instead been applied to the generation of antimicrobial properties for biomedical applications on additively manufactured structures [25]. Overall, surface-induced polymerization is often used to generate modified surface properties, such as superwettability [26], antimicrobial [27], and antifouling properties [28].

The current publication focuses on establishing a higher surface-area-to-volume-ratio of 3D-printed structures through the surface-induced polymerization of a stimulus-sensitive material. Subsequent enzyme immobilization enables the applicability in a fluid dynamically enhanced plug flow reactor. Furthermore, the stimulus-sensitivity after enzyme immobilization is investigated, and the universality of the *smart* enzyme carrier in aqueous and organic media is demonstrated to pave the way for autonomous process control and enable more efficient processes.

## 2. Results and Discussion

In the following, the results on the preparation of the here developed carriers, as well as the enzyme immobilization on surface-enhanced 3D-printed carriers are discussed. As a model reaction, biotransformations for one, the hydrolysis of butyl levulinate in aqueous medium catalyzed by Esterase 2 from *Alicyclobacillus acidocaldarius* (Est2) and secondly, the transesterification of vinyl acetate with cinnamyl alcohol in an organic solvent catalyzed by lipase B from *Candida antarctica* (CalB) were chosen. The immobilization procedure was the same for both enzymes and was performed on bare polyamide 12 (PA12) and on PAA-grafted PA12 (PA12-PAA).

PA12 is 3D-printed with an SLS (selective laser sintering) approach in the same geometry as a previously published periodic open-cell structure [12,13]. Pretreatment of the respective 3D-printed structure or particles with hydrochloric acid lead to the formation of a carboxylic acid and an amine group [29]. The carboxylic acid group is utilized to perform several steps of surface functionalization according to Figure 1, resulting in a surface-initiated polymerization of PtBA (poly(*tert*-butyl acrylate)), which is further hydrolyzed to PAA (polyacrylic acid).

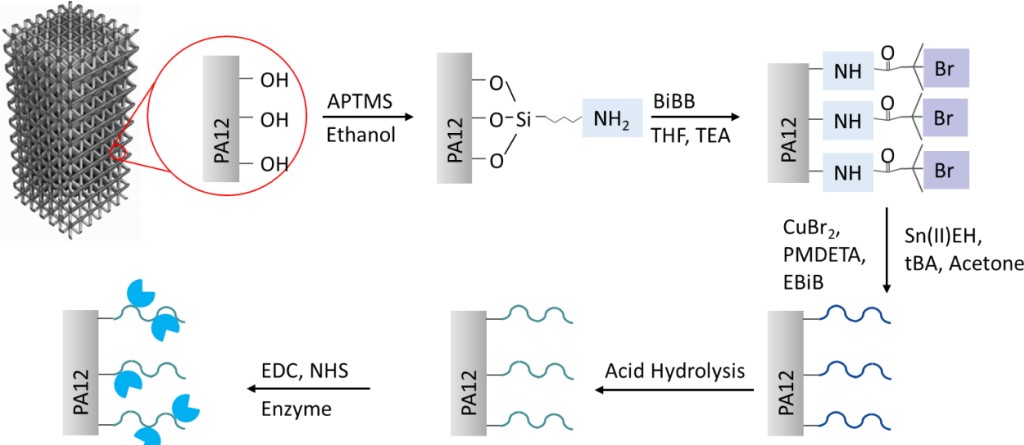

**Figure 1.** Stepwise functionalization process of the support structure made out of PA12 (Polyamide 12). First hydrolysis reaction leads to carboxylic acid and amine groups on the surface (not shown). The carboxylic acid, or rather the hydroxy group, is further utilized for subsequent functionalization steps. Successive aminosilane functionalization leads to surface-standing amine-groups, which are further reacting with the bromine-containing initiator. The initiator works as starting point for the polymerization to achieve polymers, covalently bound to the surface. After PtBA (poly(*tert*-butyl acrylate)) synthesis, acid hydrolysis is applied to convert PtBA (blue) to the stimuli-sensitive PAA (polyacrylic acid; green), where enzymes are attached via covalent EDC/NHS coupling. Abbreviations: 3-aminopropyltrimethoxysilane (APTMS), 2-bromoisobutyryl bromide (BiBB), tetrahydrofuran (THF), triethylamine (TEA), N,N,N′,N″N″-pentamethyldiethylenetriamine (PMDETA), 2-bromoisobutyrate (EBiB), tin-(II) 2-ethylhexanoate (Sn(II)EH), *tert*-butyl acrylate (tBA), 1-ethyl-3-(3-dimethylaminopropyl)carbodiimide hydrochloride (EDC), sulfo-*N*-hydroxysuccinimide (NHS).

To verify the successful surface functionalization, as well as polymerization, detailed analytics are demonstrated in the following.

### 2.1. Surface Functionalization

The determination of surface functionalization on 3D-printed structures was performed by X-ray photoelectron spectroscopy (XPS) and Fourier-transform infrared spectroscopy (FTIR), where each functionalization step was investigated. Firstly, the bare PA12 structure (Figure 2A) already contains low amounts of silicon within the 3D-printed matrix, besides the expected carbon, nitrogen, and oxygen atoms. This might be introduced by the post-modification of the additively manufactured packing, which is often performed by

sandblasting to remove powder residues [30]. No significant changes were visible after the acid treatment of PA12 (PA12-HCl); the respective atom-% (at%) is summarized in Table 1. After the aminosilane modification of the surface, the amount of silicon increased by 2.17 at%, as demonstrated in Table 1 and Figure 2B. This is indeed related to the successful functionalization of aminosilane. The subsequent covalent attachment of the bromine-containing initiator is effectively shown in Figure 2C, where characteristic peaks for Br (Br 3d) appeared. Additionally, 1.31 at% of bromine is displayed in Table 1. Eventually, as a consequence of polymerization, the whole surface composition was changed. A drastic increase in oxygen at% from 11.39 at% to 29.77 at%, which is related to the ester bond in *tert*-butyl acrylate (tBA), was measured. Additionally, no nitrogen was detectable after surface-induced polymerization, stressing the successful polymerization of tBA. Tin was implemented on the surface (Figure 2D), leading to the conclusion that the tin-containing reducing agent, which was applied throughout the polymerization reaction, seemed to be incorporated into the polymer. This could be attempted to be removed by excessive washing steps after polymerization.

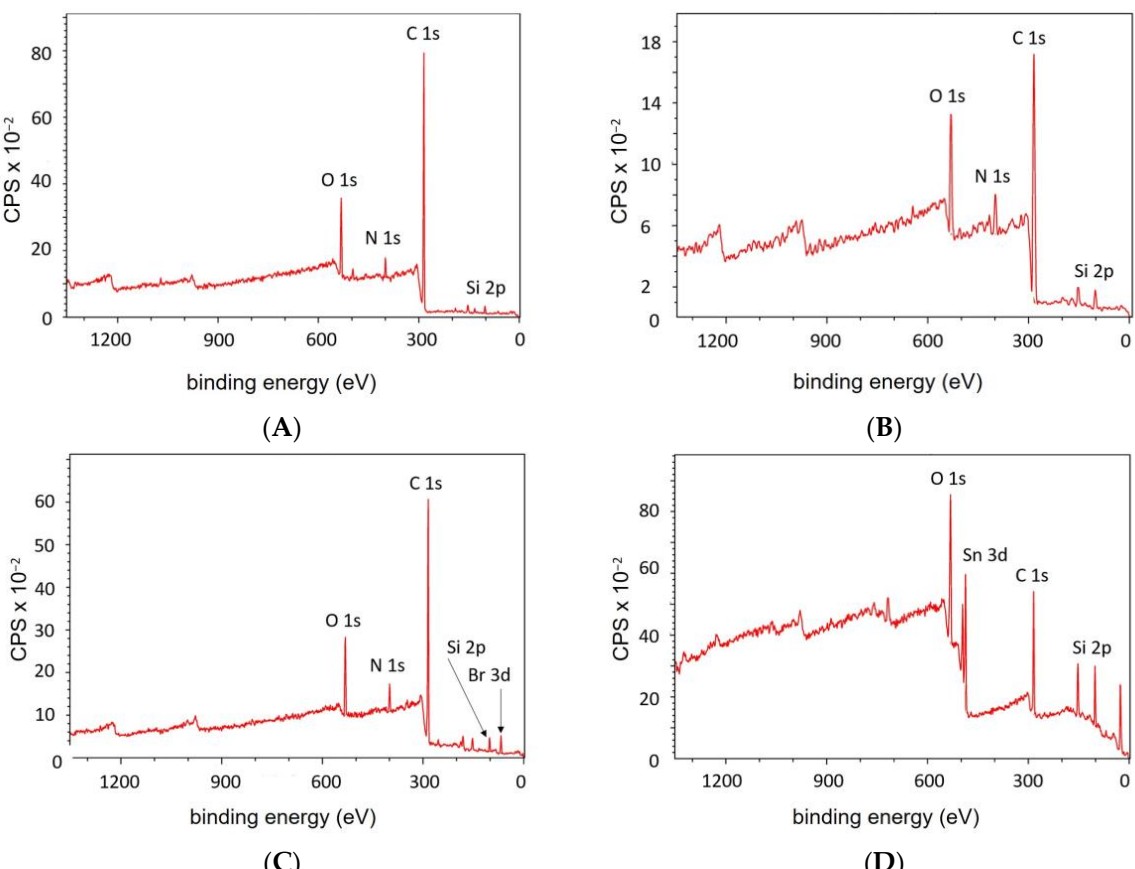

**Figure 2.** X-ray photoelectron spectroscopy (XPS) wide-scan of (**A**) PA12, (**B**) APTMS functionalized PA12, (**C**) BiBB functionalized PA12, (**D**) PtBA-grafted PA12.

**Table 1.** Atom % (at%) of various atoms on the surface of the respective structures, analyzed via XPS.

| Structure | O 1s (at%) | C 1s (at%) | N 1s (at%) | Si 2p (at%) | Br 3d (at%) | Sn 3d (at%) |
|---|---|---|---|---|---|---|
| PA12 | 11.39 | 82.22 | 4.1 | 2.3 | - | - |
| PA12-HCl | 15.36 | 75.64 | 5.43 | 3.58 | - | - |
| PA12-APTMS | 12.3 | 77.3 | 5.93 | 4.47 | - | - |
| PA12-BiBB | 10.36 | 79.42 | 5.54 | 3.37 | 1.31 | - |
| PA12-PtBA | 29.77 | 47.82 | - | 17.53 | - | 4.88 |

Overall, high measurement accuracy could be derived from the XPS data of the untreated PA12, as the deconvolution and quantification of the C 1s region showed good agreement with the literature values [31]. XPS proved the successful and covalent functionalization of PA12 surfaces, with polymerization also being confirmed by significant changes in the atom composition of the studied surface.

Similar results were obtained by FTIR, where the spectrum of PA12 and acid hydrolyzed PA12 did not show significant differences, as shown in Figure 3A. Bands at 1544 cm$^{-1}$ belonged to the amide II band (C-N stretching and CO-NH bending) [32]. Furthermore, CH$_2$-bands were also visible (2916 cm$^{-1}$ and 2847 cm$^{-1}$), in addition to the hydrogen-bonded N-H stretching at 3291 cm$^{-1}$ [32].

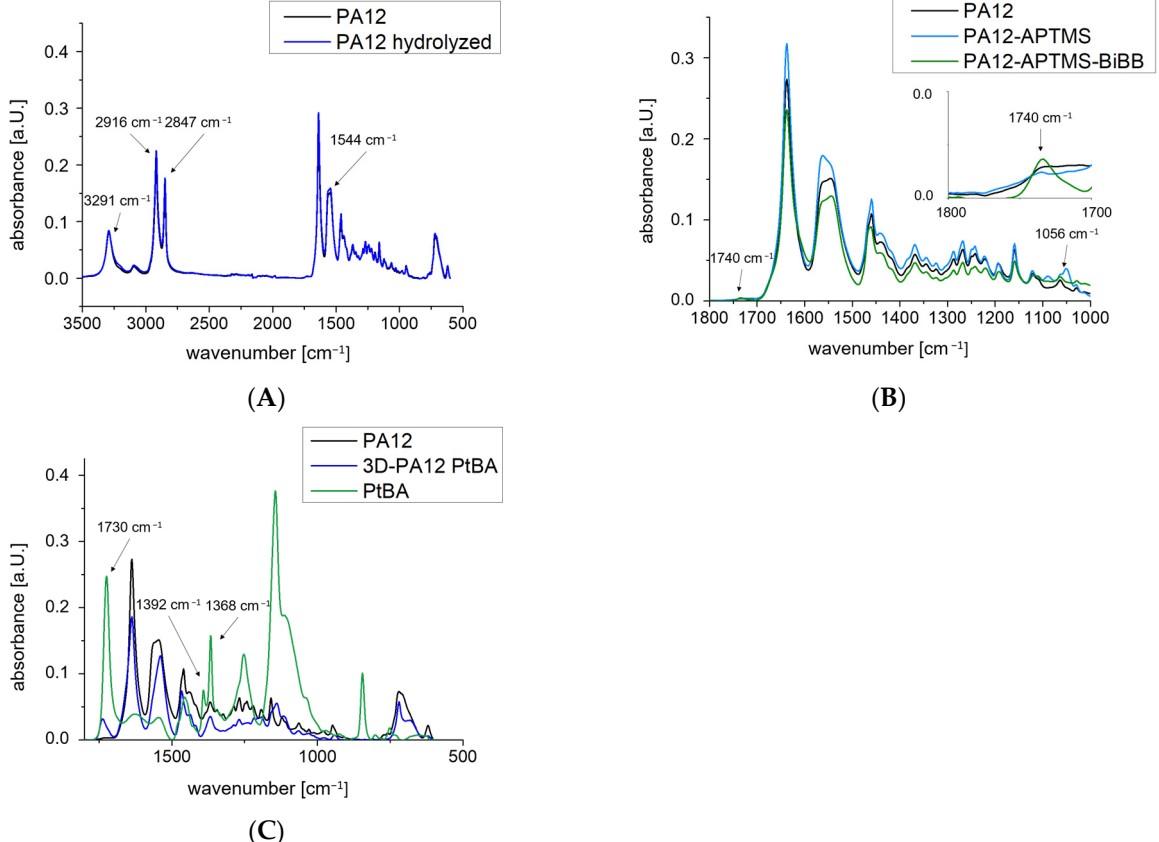

**Figure 3.** Fourier-transform infrared spectroscopy (FTIR) analysis of (**A**) PA12 and acid hydrolyzed PA12, (**B**) APTMS and BiBB functionalized PA12, (**C**) PtBA-grafted 3D-printed PA12 in comparison to PA12 and PtBA. The presented range of wavenumbers is chosen depending on the most significant changes within the spectra.

In addition, the aminosilane-modified surface led to the introduction of an additional band at 1056 cm$^{-1}$, which corresponded to the Si-O bond (Figure 3B) [33]. The successful bromine-modification is also visible in Figure 3B due to an additional band at 1740 cm$^{-1}$ that could correspond to the introduced C=O double bond. Finally, the surface-induced polymerization led to an additional band at 1730 cm$^{-1}$ (C=O stretch) and an additional doublet at 1368/1392 cm$^{-1}$ (symmetric methyl deformation mode) that correlate with the spectrum of PtBA; see Figure 3C [34]. Furthermore, to ensure the covalent binding of APTMS and BiBB, $^1$H NMR analysis was performed of the covalent APTMS-BiBB binding, where the characteristic proton shifts, such as the CH$_2$-CH$_2$ (1.66 ppm) and CH$_2$-NH-C=O (6.89 ppm), were detectable (Figure S1). In addition, the assigned protons were in good correlation with Wang et al. [35], as well as the auto-alignment by MestreNova (Ver. 14.2.3-29241-2022, Mestrelab Research S.L., Santiago de Compostela, Spain).

## 2.2. Polymerization and Topografic Characterization

Scanning electron microscopy was performed for the 3D-printed PA12 structure and the PtBA-grafted 3D-printed PA12 structure (Figure 4). Here, the structures were produced by SLS, where powder was sintered towards the desired 3D geometry. This resulted in the particle geometry still being visible on the 3D-printed structure, as demonstrated in Figure 4A. In contrast, after PtBA synthesis, the surface seemed to be covered by an additional polymer layer, leading to a smoother surface compared to the non-treated 3D-printed structure (Figure 4B). For the analysis of a successful polymerization process, the synthesis was performed in free solution and investigated by $^1$H NMR, demonstrating efficient PtBA synthesis (see Supplementary Information Figures S2 and S3).

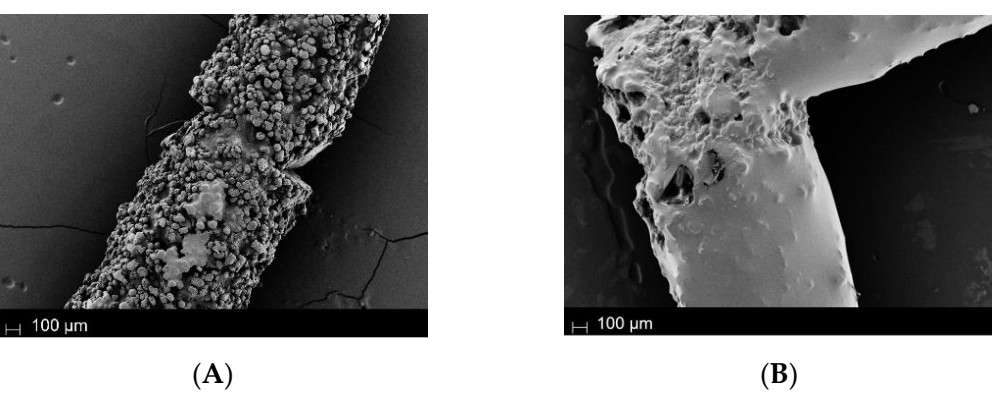

(**A**)                    (**B**)

**Figure 4.** Scanning electron microscopy (SEM) images of (**A**) PA12 3D-printed structure and (**B**) PtBA-grafted PA12 3D-printed structure. SEM images were captured with a secondary electron detector.

To demonstrate surface enhancement, or at least an increased number of surface functional groups leading to a higher number of chemical binding-points for the subsequent enzyme immobilization, a qualitative and quantitative determination of enzyme immobilization was performed. For comparison, green fluorescent protein (GFP) was immobilized on the 3D-printed PA12 structure without surface modifications, Figure 5A, and on the PAA-grafted 3D-printed PA12 structure (Figure 5B).

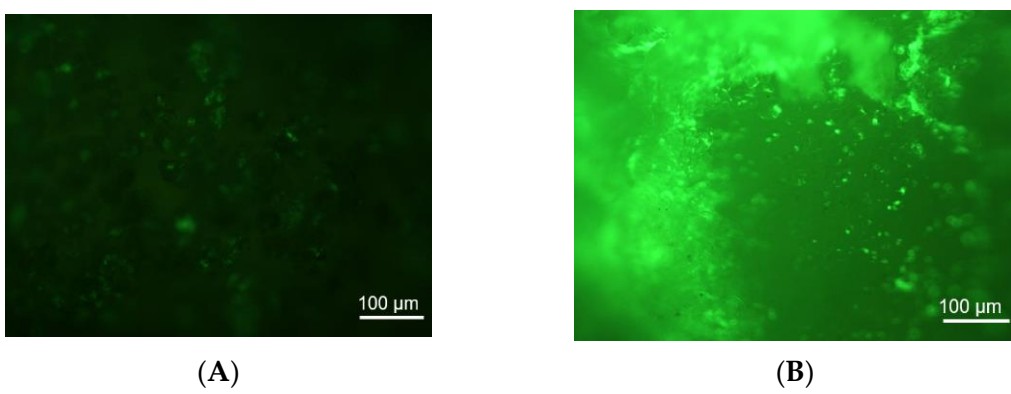

(**A**)                    (**B**)

**Figure 5.** Fluorescence microscopy images of GFP (green fluorescent protein) immobilized on (**A**) PA12 3D-printed structure and (**B**) PAA-grafted PA12 3D-printed structure.

A higher amount of immobilized GFP was detected after surface-induced polymerization, as shown in Figure 5 by change in the fluorescence intensity. Consequently, a rise in the number of surface functional groups was proven. The quantification of immobilized enzymes was performed according to Bradford [36] and proofed a six-fold higher amount on the PA12-PAA structure compared to the unmodified PA12 structure.

### 2.3. Influence of Grafted Polymer Layer on Stimuli Responsivity and Enzyme Immobilization

Due to the fact that tin ions were detectable by XPS analysis (Figure 2D), the influence of 1 mmol/L $SnCl_2$ was investigated for the two enzymes catalyzing the chosen model reactions. One is the transesterification of vinyl acetate with cinnamyl alcohol catalyzed by lipase B from *Candida antarctica* (CalB) in organic solvent. Furthermore, a hydrolysis reaction of butyl levulinate was chosen for aqueous applications, which is catalyzed by Esterase 2 from *Alicyclobacillus acidocaldarius* (Est2). To determine the influence of $Sn^{2+}$ ions in particular, high throughput screening was performed with both enzymes with and without additional ions, see Figure 6.

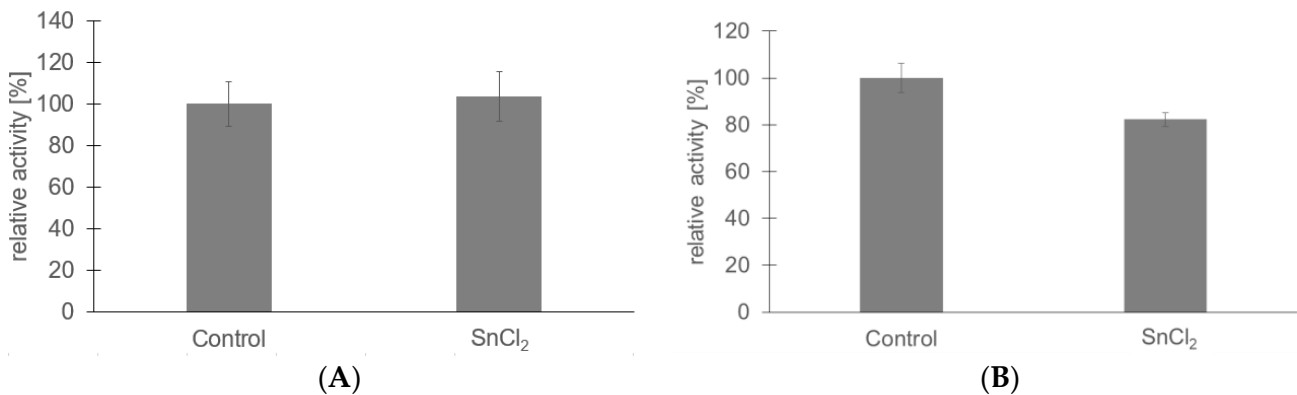

**Figure 6.** Activity of CalB (*Candida antarctica* lipase B) without additional ions (control) and with 1 mM of $SnCl_2$ (**A**), activity of Est2 without additional ions (control) and with 1 mM of $SnCl_2$ (**B**). Both enzymes were screened according to the standard activity assay in triplicates.

Overall, as Figure 6A proofs, CalB does not show significant changes in the activity because of the presence of 1 mmol/L $SnCl_2$, whereas the relative activity of Est2, in Figure 6B, decreased to 82%. Thus, only Est2 catalyzed reactions might be influenced by the tin residuals in the synthesized polymer layer.

In order to find an optimum between the maximum amount of enzyme immobilized, its highest residual activity, and a significant swelling-degree that can slow down the conversion, the dependence of the synthesis time on the respective parameters was investigated. As enzyme immobilization took place at the carboxylic acid group of PAA, which was responsible for its pH-sensitivity or hydrogen bond formation [22,37]; the swelling behavior of PAA with immobilized enzymes was investigated.

Figure 7 shows the increasing median diameter (e.g., pH 5) of the measured particles at polymer synthesis times between 0 h (unmodified) and 6 h. Significant swelling with increasing pH was detectable for all particles except the unmodified particles, proving that the synthesized polymer swells or shrinks with increasing or decreasing pH. Whether the swelling behavior resulted from stimuli-sensitive behavior or hydrogen bonds was not clarified within the scope of this publication. In addition to stimuli-sensitive behavior, hydrogen bonds also led to a shrinkage of the polymer layer since hydrogen bonds form predominantly in acidic conditions [22,38]. The results presented here, especially for a synthesis time of 6 h, show some outliers (e.g., pH 6 and 9) due to agglomeration and deagglomeration processes during the surface functionalization and polymerization steps. This could be overcome by the ultrasound treatment of particles prior to the surface functionalization steps.

Overall, the trend of synthesis time and swelling behavior, even with immobilized enzymes, was clearly demonstrated. Longer synthesis time led to an increased layer thickness. Additionally, particle size increased with higher pH values, either due to pH-sensitive behavior or due to hydration effects [22].

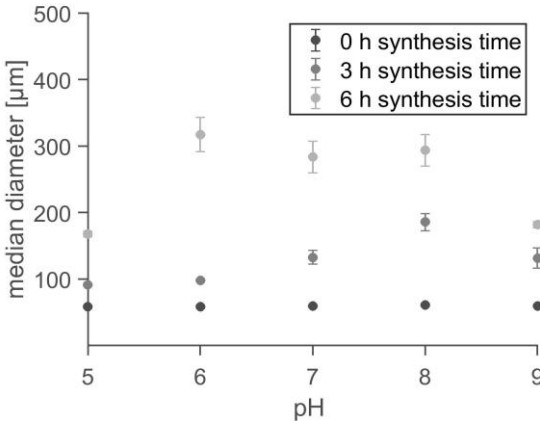

**Figure 7.** Dynamic light scattering (DLS) measurement of PA12 particles with grafted polymer and immobilized enzymes (Est2). Polymer synthesis time varies between unmodified (0 h) and 6 h. An increasing median diameter is visible with increasing synthesis time. Additionally, all particles, besides unmodified ones, proof significant swelling behavior. All measurements were conducted in triplicate.

The respective enzyme activity per $g_{support}$ was determined for several PA12-particles with a PAA-synthesis time between 0 h (unmodified) and 6 h, as shown in Figure 8.

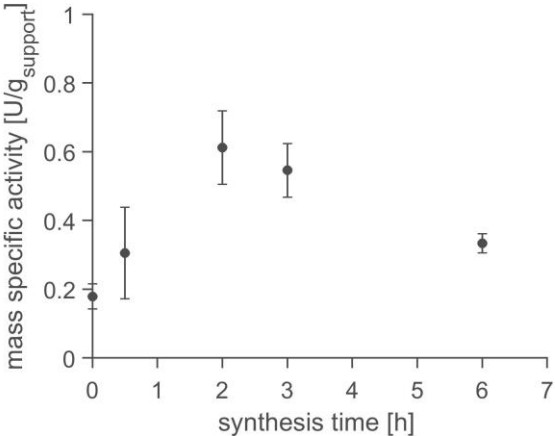

**Figure 8.** PAA-grafted PA12 particles were prepared with different synthesis time, thus different PAA layer thickness. Subsequently, enzymes (Est2) were immobilized and tested for their activity at 40 °C with the standard activity assay (5 mM *p*NP-acetate, 10% (*v/v*) acetonitrile in 163 mM PBS buffer (pH 7.0); triplicates). Increasing layer thickness up to 2 h synthesis time leads to higher enzyme activity per $g_{support}$. However, longer synthesis times led to lower enzyme activity per mass, hinting to diffusion limitation.

Shorter synthesis times (0–2 h) led to a lower layer thickness, as shown in Figure 7, so only a limited number of enzymes were immobilized. In general, longer synthesis times resulted in a greater quantity of enzymes being immobilized. Among these samples, an increase in enzyme activity is measured with increasing synthesis time, as low diffusion limitation is present. In contrast, synthesis times between 3 and 6 h result in larger polymer layers but lower enzyme activity per $g_{support}$ since diffusion limitation increases with longer synthesis times.

The highest activity per $g_{support}$ was determined for polymer-layers with 2 h synthesis time. To achieve results in regard to how the final enzyme activity is affected by the swelling state, the potential of the respective polymer layer to completely suppress the enzyme catalyzed reaction is determined.

In addition to the ability to completely stop the enzyme-catalyzed reaction, the response time of the swelling behavior also plays a crucial role in autonomous process control. Therefore, the corresponding particles were incubated at pH 7.0 for 6 measurements with incubation times between 0 and 24 h to determine the response time of the swelling behavior of the synthesized polymer (Figure 9).

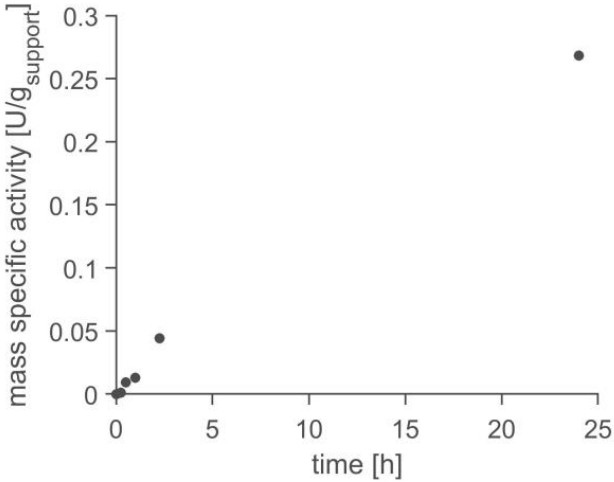

**Figure 9.** Activity of immobilized enzyme (Est2) per $g_{support}$ in dependency of the incubation time in 163 mM PBS buffer, pH 7.0. The increase in activity is constant over 24 h, leading to the conclusion that the synthesized PAA-layer has a very slow swelling response, which is to be optimized for future autonomous process control. Standard activity test was performed at 40 °C, 5 mM *p*NP-acetate in 10% (*v/v*) acetonitrile in 163 mM PBS buffer.

In this work, synthesized PAA-grafted PA12 particles proofed a very long swelling-equilibration time. Due to the increasing specific activity between 0 and 24 h, the swelling-equilibration might not be reached within 24 h. In contrast, Yang et al. found PAA equilibration times of about 10 to 13 h, depending on the cross-linking density [39], whereas PAA brushes, which, by definition, are not cross-linked, tend to have very quick adaption times, as published by Liu et al. [40]. To achieve autonomous reaction control, the response time of PAA has to be as quick as possible. The here-presented approach can be further improved by the synthesis of PAA brushes on a 3D-printed PA12 structure, leading to a high potential enzyme carrier for future application in industry.

### 2.4. Application in Batch Reactor

To prove the universal applicability of the surface-enhanced 3D-printed carrier, batch reactions were carried out for the two reference biotransformations. The hydrolysis reaction, catalyzed by Est2, allows autonomous process control by continuously changing the pH-value in the reaction system through acid production. The transesterification reactions, catalyzed by CalB, can potentially benefit from stimuli-sensitive behavior by utilizing the shift in solvent composition as a trigger for phase transition of materials such as pNiPAAm (Poly(*N*-isopropylacrylamide)) gels [41].

Overall, the immobilization procedure was the same for both enzymes and was performed on bare PA12 and on PAA-grafted PA12. The 3D-printed structures used had the same geometry and size; therefore, the following data are normalized per structure.

The immobilization of CalB on PA12-PAA leads to very low activity compared to its immobilization on PA12. Despite immobilizing a similar quantity of enzyme on PA12-PAA (1392 µg), Figure 10 demonstrates that a comparable conversion to PA12-CalB (1355 µg) cannot be achieved.

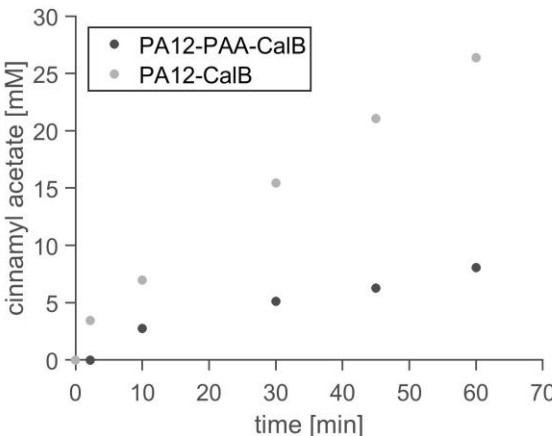

**Figure 10.** Comparison of CalB immobilized on PA12-PAA (PA12-PAA-CalB) and on PA12 (PA12-CalB) 3D-printed structure. Faster conversion was determined for PA12-CalB, even with similar amount of enzyme immobilized compared to PA12-PAA-CalB. Reaction conditions: 30 °C, in 1,4-dioxane, 300 mM cinnamyl alcohol, 600 mM vinyl acetate, 300 rpm.

CalB is known to be activated on hydrophobic surfaces [42], e.g., on PA12. In addition, covalent immobilization generally leads to reduced activities of lipases [43]. Besides the physicochemical properties of the carrier, which play a significant role in the final characteristics of the immobilized enzymes [44], diffusion limitation can also be a challenge for PAA-grafted PA12, but this can be compensated by application in a plug flow or rotating bed reactor. Also, the immobilization procedure will have an influence on the residual enzyme activity, as EDC/NHS-coupling is based on reactions with exposed lysine residues. In CalB, there are only three lysine residues surface exposed, which are located at the opposite site compared to the active center, which involves serine, histidine, and lysine [45].

Furthermore, PAA is reported to be soluble in 1,4-dioxane, depending on its molecular weight and the temperature of the solvent [46]. This could have a great influence on the resulting apparent enzyme activity under the respective reaction conditions. Figure 11 proves that there were significant changes in the surface topology of the 3D-printed PA12-PAA-CalB after application in the respective organic solvent. The exact mechanism that was triggered during the transesterification process in the organic solvent is not clear yet, but it is definitely related to the surface functionalization and polymerization process, as no similar change was observed for PA12-CalB.

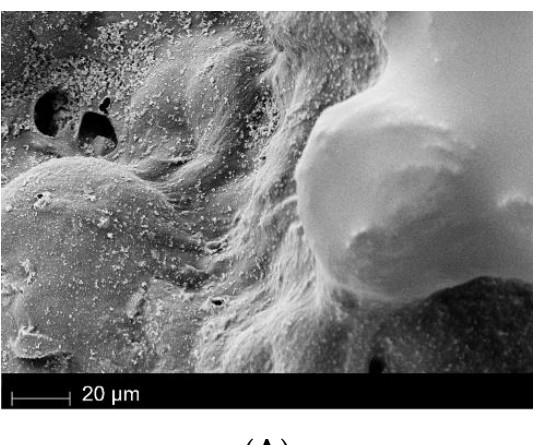

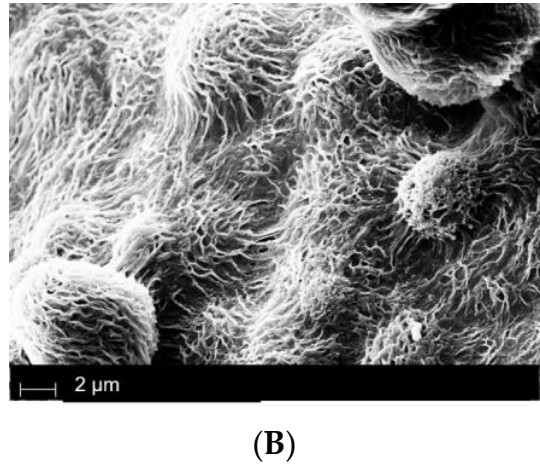

**(A)**                    **(B)**

**Figure 11.** SEM images of PA12-PAA-CalB before (**A**) and after (**B**) application in organic solvent (1,4-dioxane) for 60 min at 30 °C. SEM images were captured with a secondary electron detector.

Similarly, the Est2 catalyzed model reaction was performed in aqueous media, comparing PA12-Est2, PA12-PAA-Est2, and non-immobilized Est2 (Est2 free). The influence of the swelling behavior of the PAA-modified surface was investigated in detail. With reaction progress, the pH should decrease and lead to a shrinkage of the PAA-layer. This can significantly affect the residual activity of the immobilized enzyme by either protecting it from the environment or increasing diffusion limitation, with both leading to reduced activity. Also, the pH itself influences the residual activity of Est2, as shown in the SI (Figure S4). Both acidic and alkaline conditions lead to a reduction of the relative activity of Est2 compared to a neutral pH. The stimuli-sensitive behavior can be tracked at a pH between 5 and 9, which would result in a continuous increase in activity with an increase in pH due to the polymer swelling. Overall, this effect is only at very low specific activities, due to the pH-dependent activity of Est2. Free Est2 has the highest activity, despite being used in lower amounts than the applied 3D-printed structures with immobilized Est2. However, it did not reach the highest conversion within a 90 min reaction time (Figure 12). The pH remained stable during the reaction catalyzed by free Est2, and there was no measurable decrease in enzyme activity due to PAA shrinkage. Therefore, it is assumed that PAA did not shrink during the experiments with PA12-PAA-Est2, due to the very stable pH and the slow equilibration time, as shown in Figure 8.

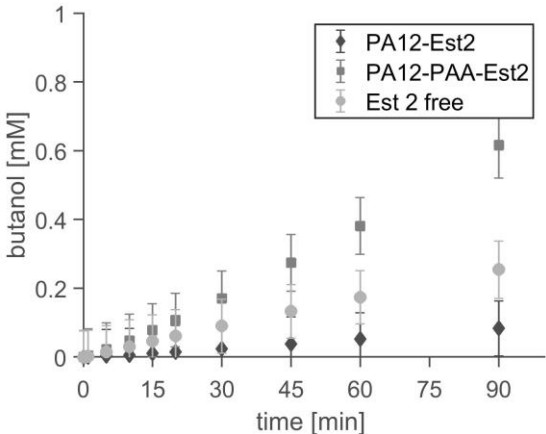

**Figure 12.** Comparison of Est2 immobilized on PA12, and PA12-PAA, as well as non-immobilized Est2 (free Est2). Faster conversion was detected for PAA-grafted PA12 structure; still, highest activity is measured for free Est2. Reaction conditions: 20 mM butyl levulinate in 10% (*v/v*) acetonitrile and 163 mM PBS buffer (pH 7.0), 40 °C, 300 rpm, respective 3D-printed structure, or 1.92 µg/mL Est2 (0.1 U/mg).

Nevertheless, Est2 immobilized on PA12-PAA (PA12-PAA-Est2) proofs significantly higher activities in contrast to PA12-Est2. Even though the immobilization of Est2 on PA12 resulted in a 1006 µg bound enzyme, only 1% residual activity was detected. In contrast, 669 µg enzyme were immobilized on PA12-PAA. Here, the number of immobilized enzymes actually contradicts the previously demonstrated surface enhancement, as no controlled polymerization was achieved, and the reproducibility of the here-applied surface-induced polymerization was low. In addition, residual tin ions, as shown in Figure 2D, should have a negative impact on the Est2 activity, as proven in Figure 6B. Nevertheless, PA12-PAA-Est2 showed, in contrast to PA12-Est2, improved residual activities of 10%. Thus, here, the physicochemical properties of the carrier seem to have a larger positive influence on the residual enzyme activity compared to the negative influence of the tin ions. Additionally, the amount and position of exposed lysine residues should be taken into account, as the immobilization procedure is based on a reaction with the surface exposed lysine residues. Overall, Est2 contains twelve lysine residues, but not all of them are surface-exposed. Furthermore, proximity to the active site is essential to finding a correlation between loss of activity and immobilization procedure. According to the Swiss-PDB Viewer (Swiss-PDB

Viewer v4.1, [47]), the Est2 (PDB: 1EVQ) possesses seven surface-accessible lysine residues. Three of these residues are in proximity of 10–25 Å of the active site, whereas the residual four lysine residues are distributed in larger distances from the active site. Therefore, there is a chance of active site hindrance by immobilization.

The recyclability of the novel enzyme carrier was evaluated for applications in buffer. Five repetitive experiments were performed, demonstrating nearly a 50% decrease in conversion between the first and second run but a consistently similar conversion for the following experiments (Figure 13).

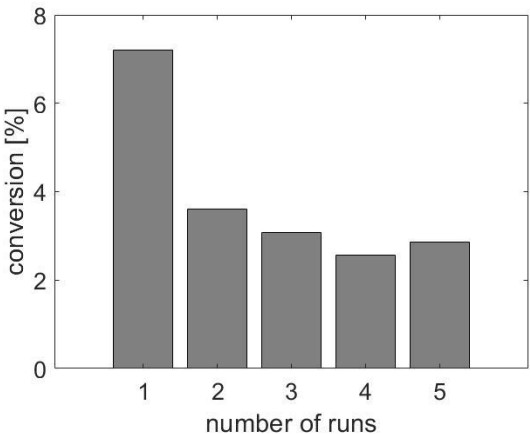

**Figure 13.** Recyclability of PAA-grafted 3D-printed PA12 structure with immobilized Est2 in aqueous media. Product concentration below detection limit after five runs. First and second run proofs showed almost 50% reduction of conversion, with stable results between runs two and five. Reaction conditions: 20 mM butyl levulinate in 10% (*v*/*v*) acetonitrile and 163 mM PBS buffer (pH 7.0), 40 °C, 300 rpm, respective 3D-printed structure.

Overall, the conversion after 1 h reaction time was very low; therefore, after five experiments, the product concentration stayed below the detection limit of 2.5% conversion. Although no enzyme leaching was detected over the course of five days, the carrier's process stability needs to be reassessed with extended reaction times to achieve higher conversion rates per run and generate additional data for subsequent runs.

## 3. Materials and Methods

### 3.1. 3D Printing of PA12 Structure

The periodic open-cell structures with a 6 mm cubic unit cell, oriented on tip, were produced with the EOS Eosint P396 and a standard PA12 polymer powder. A layer thickness of 120 μm was used. The structures were thankfully provided by the Institute of Laser and System Technologies (iLAS) from the Hamburg University of Technology (TUHH).

### 3.2. Surface Modification of PA12

Either PA12 particles from the PA12 polymer powder or 3D-printed PA12-structures were firstly hydrolyzed in 3 mol/L hydrochloric acid for 60 min at room temperature. Subsequently, material was dried after washing extensively with water and ethanol.

Amino-functionalization was conducted at 50 °C overnight with 1–2 g of particles or small-sized 3D-printed structures in a 10% (*v*/*v*) (3-aminopropyl)trimethoxysilane solution in ethanol. Overall, about 30 mmol of APTMS was used per g of particles/structure. Afterwards, particles or structures were washed with ethanol thrice and dried overnight.

Subsequent initiator-modification was performed in two Schlenk flasks that were first evacuated and back-filled with nitrogen three times. Afterwards, the amino-functionalized substrate was inserted into one flask, maintaining nitrogen flow. The second flask was filled with 20 mL dry tetrahydrofuran (THF) and 2 mL α-bromoisobutyryl bromide (ca. 16 mmol per g of particles/structure). The first flask containing the amino-functionalized substrate

was kept on ice and filled with 30 mL THF and 0.5 mL triethylamine. Subsequently, the second flask was added to the first flask dropwise under inert conditions. The reaction occurred for 30 min at 0 °C and consequently for 20 h at room temperature, with constant nitrogen flow. The initiator-modified substrates were washed three times with ethanol and were dried overnight.

### 3.3. Polymer Synthesis on Modified PA12

The polymer synthesis approach was modified from Berger et al. [48]. An inert atmosphere was necessary throughout the whole polymerization. The monomer (*tert*-butyl acrylate, tBA) was filtered over basic aluminum oxide direct before use. The composition for poly(*tert*-butyl acrylate) (PtBA) synthesis is listed in Table 2.

**Table 2.** PtBA polymer synthesis components.

| Pre-Functionalized Particles [g] or 3D-Printed Structure | ca. 1 | |
|---|---|---|
| *Tert*-butyl acrylate [mL] | 5.0 | 30 mmol |
| Cu(II)Br (catalyst) [g] | 0.0004 | 0.002 mmol |
| N″-pentamethyldiethylenetriamine (ligand) [mL] | 0.0018 | 0.009 mmol |
| ethyl 2-bromoisobutyrate (initiator) [mL] | 0.0009 | 0.006 mmol |
| acetone [mL] | 1.0 | 0.014 mmol |
| tin(II) 2-ethylhexanoate [mL] | 0.3 | 0.926 mmol |
| anisol [mL] | 2.7 | 0.025 mmol |

The components were added right before heating the reaction medium to 70 °C in a water bath for the corresponding time. Stirrer speed was set to 800 rpm. After completion, the reaction mixture was filtered, and the polymer-grafted substrate was extensively washed with acetone and methanol in alternating order and dried afterwards.

Conversion of PtBA to PAA was conducted via hydrolysis in 10% (*v/v*) methanesulfonic acid in dichloromethane (DCM) for 15 min. Afterwards, the treated substrate was washed once with DCM and twice with ethanol and finally dried overnight.

### 3.4. Enzyme Immobilization on Unmodified and PAA-Grafted PA12

Covalent immobilization on PA12 or PAA-grafted PA12 were conducted in a similar manner. PA12 was hydrolyzed for 60 min in 3 mol/L HCl before immobilization, whereas PA12-PAA was applied without additional pretreatment. EDC/NHS coupling was performed in 0.1 mol/L EDC (1-ethyl-3-(3-dimethylaminopropyl)carbodiimide) and 0.1 mol/L NHS (*N*-hydroxysuccinimide) in distilled water for 30 min at room temperature. Subsequently, the substrates were washed twice with ethanol and twice with phosphate buffer (20 mmol/L, pH 7.0), and the substrates were immersed in enzyme solution (1 mg/mL) overnight at room temperature. Finally, the immobilisate was washed thrice with 163 mmol/L PBS buffer (pH 7.0), and immobilization yield was determined according to the protein concentration determination.

#### 3.4.1. Standard Activity Assay

High-throughput activity screening was performed with *para*-nitrophenol-acetate in 163 mmol/L PBS buffer (pH 7.0). Therefore, *p*NP-acetate was dissolved in 10% (*v/v*) acetonitrile/PBS buffer (Est2), or in 10% (*v/v*) Dimethylsulfoxide/PBS buffer (CalB), leading to a final concentration of 5 mmol/L *p*NP-acetate in the reaction volume. In case of investigating ion influence, SnCl$_2$ was added in the respective amount into the reaction solution. Reactions at 70 °C were incubated for 5 min at the respective temperature before addition of the catalyst. pH-Dependent activity was determined in Britton–Robinson buffer (40 mmol/L boric acid, 40 mmol/L phosphoric acid, and 40 mmol/L acetic acid; pH values were adjusted using 3 mol/L NaOH). Additionally, 200 μL samples were taken after every 30 s for the time span of 2 min and analyzed in the iTecan multiplate reader Infinite$^®$ 200 PRO at 405 nm (Tecan Trading AG, Männedorf, Switzerland). pH-dependent activity

was determined at the isosbestic point of *p*NP, at 348 nm [49]. Due to the unknown layer thickness in multiwell plates, the extinction coefficient was determined in relation to the unknown layer thickness d of 250 µL sample. $\varepsilon_{405nm}/d = 6.0733$ mL/µmol. Est2 from *Alicyclobacillus acidocaldarius* was thankfully provided by University of Bayreuth, Lehrstuhl Biomaterialien and was prepared as previously published by Manco et al. [50]. *Candida antarctica* lipase B was provided by Novozymes A/S, Denmark (Lipozyme 7.5 mg/mL).

### 3.4.2. Determination of Protein Concentration

Measurement of protein concentration for e.g., determination of leaching behavior was conducted with Roti®Quant Solution (Carl Roth GmbH & Co. KG, Karlsruhe, Germany), based on the Bradford assay [36]. A total of 200 µL of Roti®Quant solution was mixed with 50 µL sample and incubated 5 min in the dark. Adsorption was measured at 495 nm (extinction coefficient = 0.0056 mL $\times$ µg$^{-1}$ $\times$ cm$^{-1}$). The calibration was performed with Bovine Serum Albumin in the respective concentrations.

### 3.4.3. Model Reaction in Aqueous Media

Esterase 2 from *Alicyclobacillus acidocaldarius* was applied in the hydrolysis reaction of butyl levulinate and water to butanol and levulinic acid in 163 mM PBS buffer (pH 7.0), containing 10% (*v/v*) acetonitrile. The reaction was performed in batch mode in a thermovessel at a constant temperature of 40 °C.

Additionally, 500 µL samples were taken at the respective time point, and duplicates of 100 µL each applied for extraction in 200 µL ethyl acetate. The organic solvent fraction was injected to gas chromatography (GC, HP 6890 Series, Hewlett-Packard, Palo Alto, CA, USA) equipped with an OPTIMA FFAPplus column (30 m $\times$ 0.25 mm $\times$ 0.25 µm) and FID detector. The following temperature program was applied with H$_2$ as carrier gas: 45 °C for 5 min, heating to 200 °C at 30 °C/min, and holding for 2 min. The applied pressure was 1.1 bar, with a split ratio of 50:1, 1 µL injection volume, and injector and detector temperatures of 250 °C. Typical retention times were 1.8 min for acetonitrile, 3.6 min for butanol, and 9.3 min for butyl levulinate. Levulinic acid was not detectable.

The negative control revealed no detectable autohydrolysis.

### 3.4.4. Model Reaction in Organic Media

Lipase B from *Candida antarctica* was applied for transesterification of cinnamyl alcohol and vinyl acetate to cinnamyl acetate and vinyl alcohol. Vinyl alcohol is instable and thus decays quickly. The reaction was performed in 1,4-dioxane, applying 300 mmol/L cinnamyl alcohol and 600 mmol/L vinyl acetate. At the respective timepoints, samples of about 500 µL were taken, and applied to GC (Agilent 7890A, Agilent Technologies, Santa Clara, CA, USA) equipped with an Agilent HP-5 column (30 m $\times$ 0.25 mm $\times$ 0.25 µm) and FID detector. The following temperature program was used with helium as carrier gas: 50 °C for 1 min, heating to 65 °C at 5 °C/min and additionally to 275 °C with 30 °C/min. The applied pressure was 1.1 bar with a split ratio of 20:1, an injection volume of 1 µL, and injector and detector temperatures of 300 °C and 340 °C. Typical retention times were 2.1 min for vinyl acetate, 8.4 min for cinnamyl alcohol, and 9.1 min for cinnamyl acetate.

### 3.5. Analytical Methods

Analytical methods that were applied in the current publication are described in the subsequent chapters.

### 3.5.1. Proton Nuclear Magnetic Resonance

$^1$H-NMR was applied for determination of covalent fusion of APTMS and BiBB, as previously described by Wang et al. [35]. Therefore, a mixture of BiBB (1.71 mL) and dry THF (30 mL) was added dropwise into the solution of 2.352 mL APTMS, 3.86 mL TEA and 170 mL dry THF, which was kept on ice. After 30 min on ice, the reaction proceeded at room temperature overnight. The precipitate was filtered off using a frit funnel, and

a yellowish oil was obtained after removal of the solvent via rotary evaporation. The resulting yellowish oil was dissolved in 40 mL DCM and subsequently washed with a saturated sodium bicarbonate solution and cold water, four times each. The DCM layer was dried with anhydrous calcium chloride, and after removal of the solvent, the final product was a colorless viscous oil, which was analyzed via NMR.

All measurements were conducted in 0.7 mL deuterated chloroform, with 20 μL sample and 10 μL dimethylformamide added as an internal standard. The measurements were performed in 5 mm O.D. sample tubes using the Bruker Advance I 500 (AV500; Bruker BioSpin MRI GmbH, Ettlingen, Germany) with 500.13 MHz. The obtained spectra were investigated via Bruker TopSpin v2.1, if not stated differently. Further details are given in the Supplementary Information.

### 3.5.2. Fourier-Transform Infrared Spectroscopy

FTIR measurements were performed via attenuated total reflectance (ATR), on a Bruker Vertex 80 (Billerica, MA, USA), in the range of 600 to 4000 cm$^{-1}$. As analysis software OPUS 8.5 (Bruker, Billerica, MA, USA) was used. Prior to analyzing, samples were dried overnight.

### 3.5.3. X-ray Photoelectron Spectroscopy

XPS measurements were performed using a KRATOS AXIS Ultra DLD (Kratos Analytical, Manchester, United Kingdom) equipped with a monochromatic Al K$\alpha$ anode working at 15 kV (225 W). The samples were previously dried in a vacuum oven for at least one night and subsequently introduced into a high-vacuum field before analysis via the photoelectron spectrometer. For the survey spectra, a pass energy of 160 eV was used, while for region spectra, the pass energy was 40 eV. The investigated area was $700 \times 300$ μm. For all of the samples, charge neutralization was necessary. The evaluation and validation of the data were carried out with the software CASA-XPS version 2.3.24. Calibration of the spectra was done by adjusting the C1s signal to 284.5 eV. For deconvolution of the region files, background subtraction (U 2 Tougaard or Shirley) was performed before calculation.

### 3.5.4. Fluorescence Microscopy

The Nikon Eclipse 80i (Tokyo, Japan) was applied to detect emitted fluorescence by green fluorescent protein (GFP), which was immobilized on several different carriers, as stated above. The fluorophores present in the sample were excited by a mercury vapor lamp at 450–495 nm. The emission light was then passed through a 520 nm filter, and a qualitative comparison of results was achieved.

### 3.5.5. Scanning Electron Microscopy

Surface studies were performed using the Leo Gemini 1530 SEM device (Zeiss, Oberkochen, Germany). Beforehand, samples were dried overnight and sputtered with gold for 60 s. The measurements were conducted applying 10 kV electron high tension, with a field emission cathode as source and secondary electron or in-lens detector. Finally, SEM images were analyzed using smartSEM® software (Zeiss, Oberkochen, Germany).

## 4. Conclusions

To place additively manufactured structures as a competitive immobilization matrix to conventional carriers, surface enhancement is essential. Surface-enhanced 3D-printed structures can even be one step ahead by choosing stimuli-sensitive polymers as a surface modification; not only the number of enzymes per reaction volume can be increased but also autonomous process control can be envisioned. The application of surface-induced polymerization on 3D-printed structures was successfully conducted and analyzed by XPS, SEM, and fluorescence microscopy. When subsequently used as enzyme carrier, a sixfold amount of immobilized enzyme was proven. It was observed that the pH-dependent swelling behavior of the synthesized polyacrylic acid was still present when enzymes

were immobilized within, which raised the prospect of autonomous process control of the surface-enhanced 3D-printed enzyme carrier. However, there were significant drawbacks in the form of a slow response time, most likely due to the highly cross-linked polymer, which is one of the parameters to be optimized in future research. In addition, the synthesis time of the polyacrylic acid layer had a major influence on the layer thickness and the mass-specific enzyme activity per mass of support. The highest enzyme activity of 0.61 U per $g_{support}$ was achieved after 2 h of synthesis time. Subsequently, the novel immobilization support was tested for its applicability in organic and aqueous media, leading to promising results in buffer. Finally, the process stability was tested, showing no enzyme leaching but limited recyclability. Overall, this work demonstrated a proof-of-concept in autonomous process control by stimuli-sensitive material on 3D-printed carriers in biocatalysis.

**Supplementary Materials:** The following supporting information can be downloaded at: https://www.mdpi.com/article/10.3390/catal13071130/s1, Figure S1: $^1$H NMR spectra of synthesized and purified APTMS-BiBB. Assignment via MestreNova ver. 14.2.3-29241-2022 (Mestrelab Research S.L., Spain); Figure S2: $^1$H NMR spectra of polymerization medium before polymerization initiation. The respective shifts, used for determination of monomer conversion are marked and correspond to the protons at the C=C double bond, which reacts throughout polymer synthesis. The internal standard dimethylformamide at 7.9 ppm is marked and its integral set to 1. The respective monomer bands are marked and displayed in correlation to the internal standard; Figure S3: $^1$H NMR spectra of polymerization medium after successful polymerization (120 min). The respective bands, used for determination of monomer are marked and correspond to the protons at the C=C double bond, which reacts throughout polymer synthesis. The internal standard dimethylformamide at 7.9 ppm is marked and its integral set to 1. The respective monomer bands are marked and displayed in correlation to the internal standard; Figure S4: pH-dependent activity of Est2. Reaction conditions: 0.24 µg/mL Est2 were applied with 5 mM *p*NP-acetate in Britton-Robinson buffer of the respective pH. Activity was determined at 348 nm. Highest activity at pH 7 with decreasing trend towards acidic and alkaline solutions. This is comparable to previously published data by Manco et al. [50]. Refences [35,50] are cited in the Supplementary Materials.

**Author Contributions:** Conceptualization, D.E., A.W.H.D. and A.L.; investigation, D.E., N.S., S.K. and A.K.; resources, A.L.; writing—original draft preparation, D.E.; writing—review and editing, N.S., A.W.H.D. and A.L.; project administration, D.E.; funding acquisition, A.L. All authors have read and agreed to the published version of the manuscript.

**Funding:** This project is part of the I3 program "Smart Reactors" of the TUHH and the Hamburg Ministry of Science, Research and Equality (BWFG). Publishing fees supported by Funding Programme Open Access Publishing of Hamburg University of Technology (TUHH). The authors gratefully acknowledge the financial support.

**Data Availability Statement:** Not applicable.

**Acknowledgments:** We gratefully thank the institute of Technical and Macromolecular Chemistry, University of Hamburg, especially Paulina Weidmann (working group G. Luinstra), for their continuous assistance in polymerization techniques and analyses. Furthermore, we thank the University of Bayreuth. Lehrstuhl Biomaterialien, for regularly providing the Est2.

**Conflicts of Interest:** The authors declare no conflict of interest.

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
