# Peer review of "Polymer-Grafted 3D-Printed Material for Enzyme Immobilization—Designing a Smart Enzyme Carrier"

_catalysts, doi:10.3390/catal13071130_

Round 1
Reviewer 1 Report
Interesting research on the enzyme immobilization matrices development, however, some verification is needed to improve the quality of the manuscript as follows:
What is the advantages of the 3D-printed matrices compare to another porous material for enzyme immobilization?3D print resolution maybe just in hundreds micrometer, whereas natural membrane or synthetic structure such as cryogel have smaller pore size, thus higher surface area.
What is the smart enzyme supporting material mean in this research? which results showed the smart properties?
What is the reason to use the Polyamide as matrices structure? it is biocompatible? usually natural polymer for enzyme immobilization provide better compatibility, resulting better performance to maintain the reusability in the immobilized enzyme application.
Enzyme used usually in Unit per ml or per mg. The use of mg/ml for enzyme could be confusing for future researchers to replicate the procedure, since the activity of each enzyme could be different in different brand, or even bath or production.
What it the reason to use several steps in enzyme immobilization showed in fig. 1? it seem after you have the amine group in the second steps, you can add cross linker such as glutaraldehyde to make covalent binding between the amine group in the matrices and enzymes.
From the Fig 12, it can be see that the activity of the enzyme drop with the reusability, mainly from the first to second use. Maybe some enzyme in the matrices surface just physical absorption instead of covalent bonding. Usually covalent bonding is strong enough to maintain the enzyme activity during the reusability of immobilized enzyme.
Author Response
Many thanks for your valuable comments.
The responses are attached.

Reviewer 2 Report
The authors reported an interesting work about grafting polymer on 3D printing object for enzyme immobilization. The work is well organized, and the novelty is high. However, the introduction and reference need improvement and some experiments need to be better explained. Therefore, the following revision must be done before publication.
1. Apart from using the XPS to show the surface grafting, FTIR could also be used to further prove grafting.
2. The Sn cannot be removed from polymer from Figure 2d, which is uncommon according to previous reported ATRP method. Is that possible for authors to wash their materials more thoroughly to see if the Sn can be removed or not?
3. Figure 5, it is not very accurate to get the layer thickness from top view. It is better to rotate the stage and see the cross section.
4. The introduction of 3d printing could be further expand in order to give authors a better understanding. More advantages of 3D printing could be introduced. Such as low temperature, energy saving, large materials scope, etc. The following recent examples about 3D printing should be cited (https://doi.org/10.1039/D1PY00705J; https://doi.org/10.1021/acsapm.9b00140) .
5. The characterization of acid treated PA-12 is missing. Please use FTIR/XPS to show the chemical structure.
Author Response

(The authors gave the same response as above.)

Reviewer 3 Report
Immobilization of enzymes is one key aspect for successful application of biocatalysis in industrial processes. The current publication focusses on establishing a higher surface area-to-volume-ratio of 3D-printed structures through surface-induced polymerization of a stimulus-sensitive material.
The highest enzyme activity of per gsupport was achieved. The novel immobilization support was tested for its applicability in organic and aqueous media, leading to promising results in buffer. Process stability was tested, showing no enzyme leaching.This provided evidence that autonomous process control by stimuli-sensitive material on 3D-printed carriers is possible in biocatalysis.there were interested to all the reader.
There are minor errors in the manuscript.Some suggestions are as follows:
Line 239: “o min” should be changed “o h”, or shouled be deleted “min”
Line 375,line 408,line 410: “M” should be changed to “mol·L-1 , the same errors in the manuscript such as “mM” should be changed to “mmol·L-1”
Line 570 and 632 of references: the name of Journal “Int J Adv Manuf Technol” should be changed to “Int. J. Adv. Manuf. Technol.”. “J Polym Res”should be changed to “J. Polym. Res.”
Author Response

(The authors gave the same response as above.)

Reviewer 4 Report
The manuscript of Eixenberger et al, entitled “Polymer-grafted 3D-printed Material for Enzyme Immobilization – Designing a Smart Enzyme Carrier” is a report on development polymer grafted 3D-printed support for enzyme immobilization. The surface treatment of an open cell 3D-printed polyamide 12 by sequential HCl hydrolysis, APTMS-treatment, 2-bromoisobutyryl bromide acylation, tin-(II) 2-ethylhexanoate initiated reductive poly(tert-butyl acrylate)) synthesis followed by acid hydrolysis resulted in stimuli-sensitive polyacrylic acid polymer coating layer of μm-thickness being capable of efficiently immobilize enzymes by EDC chemistry. The paper combining the benefits of 3D-printing with the enhanced enzyme binding capacity due to grafting with a proper stimuli-sensitive polymeric matrix is of real scientific interest.
The manuscript is a well-written paper using proper analytical tools – such as XPS, FTIR, SEM, fluorescence microscopy after GFP-binding, DLS – for characterization of the changes on the 3D-printed support surface.
Notes
1) Please name the enzyme(s) selected of immobilization in the Abstract and in Sub-section 2.3 (neither in text nor in Figures 7,8,9 the name of enzyme(s) is disclosed)! It is important to know even at the evaluation of the immobilization results which enzyme(s) was(were) studied. Add the information to Abstract, first sentences of Sub-section 2.1, legends of Fig. 8, 9.
2) Another suggestion is to move the first two paragraphs of second sub-section 2.3 (sub-section2.4) as the general introduction of Section 2. Results and Discussion.
3) Renumber “2.3 Application in Batch Reactor” as 2.4.
4) Figure 61 should be Figure 11 (line 320 and Figure caption).
5) Figure 72 should be Figure 12 (line 338 and Figure caption).
6) Figure 83 should be Figure 13 (Figure caption). Please name the enzyme in caption.
7) Speculation on the reason of decreased enzyme activities should be extended on analysis of the distribution of surface exposed Lys residues available for EDC chemistry of the two enzymes as compared to the active-site entrance.
8) In addition to Figure 8, another figure depicting the specific activity of the bound enzyme amount would be highly informative. The Reviewer estimates, that the specific activity maximum of the bound enzyme content is at a lower synthesis time (polymer thickness).
9) To evaluate the pH-responsive behavior, the pH-profiles of the enzymes should also be determined (taken into account).
10) Recyclability study in the flow reactor with a better test substrate would be desirable (e.g. p-NP acetate).
11) Since the amount of monomer in pre polymerization solution and post-polymerization solution are measurable (Supplementary information), it is possible theoretically to estimate the amount of polymer on the surface (missing amount) as a function of reaction time. It would be better to characterize the polymer layer by the amount of bound monomer rather than by reaction time.
12) Materials and Methods: disclose information on the enzymes (source, quality, etc.)
13) Sub-section 5.2: define temperature for HCl hydrolysis and support : reagent ratios in APTEOS silanization and α-bromoisobutyryl bromide acylation reactions!
Minor notes
- Line 124: … Fourier-transform … (name!)
Since the topic is of real interest, I recommend the acceptance of the paper after thorough addressing all of the issues listed above in a major revision.
Only a few typos shoud be corrected.
Author Response
Thank you very much for your valuable comments to further improve the manuscript. The changes are adapted as stated in the appended document.

Round 2
Reviewer 2 Report
The authors failed to address many my questions. I cannot agree to publish this paper at current form.
1. The authors should not use the track changes, instead, should highlight all the changes. Current revised manuscript is a mess. For example, Figure 3, I cannot read the FTIR data at all because the number and the figure are not aligned and very messy.
2. The FTIR characterized peak should be labeled instead of giving the raw data.
3. HNMR data should also be discussed instead of just saying " as demonstrated in the Supplementary Information"
4. For my previous question 2, the authors did not perform extra experiments to wash out the impurities in the polymer. Using the pure materials to do the application should be the very basic scientific common sense. How do you know the Sn will not influence your application? Can you do the control experiments to show that Sn has no influence on your application?
5. For my previous question 3, the authors still did not any changes. It is totally wrong that use a SEM top view to get the layer thickness, even for estimation. Publishing this wrong data will mislead readers and bring bad reputation to the authors and the journal. If the authors cannot do the experiments I asked, just delete this data and the claim.
Author Response
Thank you again, for your valuable input. The changes are adapted as stated in the document.
